# Characterization of Endothelial Cell Subclusters in Localized Scleroderma Skin with Single-Cell RNA Sequencing Identifies NOTCH Signaling Pathway

**DOI:** 10.3390/ijms251910473

**Published:** 2024-09-28

**Authors:** Theresa Hutchins, Anwesha Sanyal, Deren Esencan, Robert Lafyatis, Heidi Jacobe, Kathryn S. Torok

**Affiliations:** 1Department of Pediatrics (Rheumatology), University of Pittsburgh, Pittsburgh, PA 15224, USA; hutchinstr@upmc.edu (T.H.); sanyala@upmc.edu (A.S.); esencand2@upmc.edu (D.E.); 2Division of Rheumatology and Clinical Immunology, University of Pittsburgh, Pittsburgh, PA 15261, USA; lafyatisra@upmc.edu; 3Department of Dermatology, University of Texas Southwestern, Dallas, TX 75390, USA

**Keywords:** localized scleroderma, morphea, single-cell RNA sequencing, endothelial cells, skin, JAG-NOTCH signaling, SELE, XIST, IL6, IL33

## Abstract

Localized scleroderma (LS) is an autoimmune disease characterized by inflammation and fibrosis, leading to severe cutaneous manifestations such as skin hardening, tightness, discoloration, and other textural changes that may result in disability. While LS shares similar histopathologic features and immune-fibroblast interactions with systemic sclerosis (SSc), its molecular mechanisms remain understudied. Endothelial cells (EC) are known to play a crucial role in SSc but have not been investigated in LS. Single-cell RNA sequencing (scRNA-seq) now allows for detailed examination of this cell type in the primary organ of interest for scleroderma, the skin. In this study, we analyzed skin-isolated cells from 27 LS patients (pediatric and adult) and 17 healthy controls using scRNA-seq. Given the known role of EC damage as an initial event in SSc and the histologic and clinical skin similarities to LS, we focused primarily on endothelial cells. Our analysis identified eight endothelial subclusters within the dataset, encompassing both disease and healthy samples. Interaction analysis revealed that signaling from diseased endothelial cells was predicted to promote fibrosis through *SELE* interaction with *FGFBP1* and other target genes. We also observed high levels of *JAG* in arterial endothelial cells and *NOTCH* in capillary endothelial cells, indicating the activation of a signaling pathway potentially responsible for epidermal abnormalities and contributing to LS pathogenesis. In summary, our scRNA-seq analysis identified potential disease-propagating endothelial cell clusters with upregulated pathways in LS skin, highlighting their importance in disease progression.

## 1. Introduction

Localized scleroderma (LS), also known as morphea, is a rare autoimmune disease characterized by inflammatory-driven fibrosis. Histologically, LS is indistinguishable from systemic sclerosis (SSc) due to dense collagen deposition within the dermis and underlying connective tissue, with intermixed areas of inflammatory infiltrate, particularly peri-vascular in the deep dermis and subcutis. LS affects both the skin and underlying connective tissue in adults and children, with an annual incidence of 0.4 to 2.7 cases per 100,000 people [1]. This fibrosis in deep connective tissues can result in joint contractures; hemiatrophy of the trunk, limbs, and face; and disability in children, significantly impacting growth, with the average age of onset being 8 years old [1].

In systemic sclerosis (SSc), excessive extracellular matrix deposition in the skin is the result of initial tissue damage at the surface of endothelia (vascular). Key processes involving endothelial cells, such as endothelial cell injury, flawed angiogenesis, faulty vasculogenesis, and the endothelial-to-mesenchymal cell transformation are thought to initiate the disease process in SSc [2]. However, these processes have not yet been studied in LS [3], making it an under-investigated area.

Vasculopathy is often considered to be the initial event in SSc, manifesting as Raynaud’s phenomenon often before any other fibrotic symptoms such as skin hardening onset [4]. This early endothelial cell damage and activation lead to defective angiogenesis and vasculogenesis, often resulting in endothelial-to-mesenchymal cell transition, fibroblast activation, and subsequent fibrosis [5,6,7]. Given the similarities in the dermatopathology and the clinical skin presentation of LS and SSc, we aimed to characterize the endothelial cells (EC) in LS patients and investigate potential pathogenic cell types and aberrant interactions with immune cells and/or fibroblasts. 

Perivascular immune infiltrate, characterized as lymphoplasmacytic with mononuclear cells, is a key shared histological feature between LS and SSc [8], appearing soon after endothelial cell injury in SSc [9]. Elevated levels of circulating von Willebrand factor (vWF), endothelin-1, dead endothelial cells, and soluble JAM-1 in the peripheral blood [10,11,12,13,14,15] are indicative of endothelial cell damage and activation in SSc. Studies have shown increased expression of genes encoding vWF, tissue factor (F3), ephrin A1, and ET-1 in stimulated endothelial cells from SSc [12], highlighting the significance of endothelial cell activation and damage in the disease process. These vascular peripheral markers have not been as thoroughly studied in LS [16].

Additionally, previous research has demonstrated a high level of endothelial cell apoptosis in SSc skin, linking vascular injury to fibrosis. Earlier microarray studies have shown upregulation of vascular genes such as *SELP*, *ANGPT2*, *ICAM2*, and *VWF* in SSc skin, correlating with a high modified Rodnan Skin score (mRSS) indicative of skin fibrosis [17,18]. Single-cell RNA sequencing (scRNA-seq) in SSc skin with a focus on EC populations demonstrated high expression of perlecan (*HSGP2*) and apelin receptor (*APLNR*), which are associated with NOTCH pathway regulation [19]. 

Following the initial trigger, endothelial cells are believed to engage in crosstalk with other cell types, including immune cells, fibroblasts, and smooth muscles, promoting myofibroblasts to ultimately drive fibrosis [20]. In SSc, increased cytokine production, particularly IL-33, at sites of injury and onset has shown to enhance endothelial cell activity, permeability, and angiogenesis [21], likely through recruitment and stimulation of immune cells, fibroblasts, and myofibroblast expressing ST2 [22]. Elevated serum levels and abnormal expression of IL-33 have been detected in skin of SSc patients in early stages of the disease [23,24,25]. Additionally, IL-6 activation of endothelial cells [26] can lead to vascular leakage, leukocyte recruitment, and blocked blood flow [27,28], suggesting IL-6–mediated signaling in vascular endothelial cells contribute to damage in certain disease processes [27,28]. 

Given the similarity in the disease progression and histopathology between LS and SSc, we hypothesize that there is high heterogeneity in the endothelial cell population within LS. Furthermore, LS ECs may exhibit overlapping clusters and activity similar to those in SSc. This study aims to characterize the endothelial cell population within LS subjects, encompassing both pediatric and adult-onset LS, the various LS clinical subtypes (linear, generalized, circumscribed), using scRNA seq of the skin, comparing them to healthy controls (in the study) and published clusters of EC cells in SSc. In addition to characterizing LS EC cells, we analyzed their potential for pathogenesis in silico through cell-communication software analyses.

## 2. Results

### 2.1. Transcriptomic Evaluation of Endothelial Cells from Both LS and Healthy Control Samples Identified Higher Proportion of LS Samples in the Endothelial Cell Population

Skin biopsies from 27 LS patients and 17 age, sex, and race-matched healthy controls were enzymatically digested and were processed via our scRNA-sequencing analysis pipeline (see Section 4: Methods) [29,30]. The subjects from whom samples were analyzed included 13 LS adults, 11 healthy adults, 14 LS pediatric patients and 6 healthy children, with a female:male ratio of 5:2 (Appendix A). CellRanger version 7.0.1 was used to create matrices for each sample post single-cell sequencing from fastq files to aggregate and import the data into R for further Seurat analysis. We obtained a total of 108,239 cells, with 40,715 healthy and 67,524 LS cells. The total cell count was found to be equally split between pediatric and adult patients. 

Uniform Manifold Approximation and Projection (UMAP) identified a total of 14 main cell clusters using established canonical marker genes for the entire dataset (Figure 1A). Keratinocytes were identified using *HBB*, *HBA1*, and *HBA2;* endothelial cells (ECs) were identified using primarily *ACKR1*, *PECAM1*, *CLDN5* [31], *CCL21* [32], and *PLVAP* [33]; basal keratinocytes identified with *COL17A1*, *LAMB3*, *SYTB*; suprabasal keratinocytes identified with *LGALS7*, *SFN*, *DMKN*, *LY6D*, and *NRARP* [34]; fibroblasts were identified with *SFRP2*, *DCN* and *COL1A1* [34,35]; smooth muscle cells were identified with *ACTA2*, *TAGLN* [34,36], and *MYL9* [34,37]; monocytes/macrophages were identified with *IL1B* [38], *CD163* [39], and *CD86* [40]; T cells were identified with *CD3E*, *CCL5*, and *CD2* [41]; granular keratinocytes were identified with *IVL*, *SERPINA12* [42], and *KRT2*; eccrine glands were identified with *PIP*, *MUCL1*, and *AQP5* [43]; melanocytes/neuronal cells were identified with *S100B* [44], *MPZ*, *PMP2*, and *PMP22*; follicular keratinocytes were identified with *S100P*, *KRT6B*, *KRT6A*, and *CLIC3*; B cells were identified with *IGKC*, *CD79A* [45], and *IGHD*; and mast cells were identified with *TPSAB1* [36,46], *TPSB2*, *GATA2*, and *CTSG* (Figure 1C).

LS had a higher proportion of most cell types; notably, endothelial cells, keratinocytes, smooth muscle cells, and several immune cells such as T cells, B cells, and macrophages (Figure 1B). Since there was a higher number of LS samples, all samples were normalized with Harmony, and data were analyzed as cell proportions rather than absolute cell numbers to identify cluster prevalence per disease state. Notably, 70% of the endothelial cells were derived from LS samples compared with 30% from healthy controls. Endothelial cell canonical markers were applied to feature plots to validate expression in this population (Figure 1D). Extensive endothelial vs. all cell types DEGs can be found in the volcano plot in Appendix A.

### 2.2. Unique Subclusters of Endothelial Cells in LS Were Identified and Their Differentially Expressed Genes Were Significant When Compared with Healthy Controls

Unsupervised clustering was applied to the isolated endothelial cell (EC) object, and eight different subtypes resulting from this were visualized using UMAP (Figure 2A and Appendix A). Differentially expressed gene (DEG) analysis among the subclusters was used in tandem with established endothelial cell marker genes to identify and annotate each cluster. Arterial ECs were identified using *SEMA3G*, *GJA4*, *HEY1* [47], *FBLN5* [48] and *SELE* [47]. Venous ECs were identified with *ACKR1* [47], *CLU*, *VWF*, *CCL14* [47], and *PECAM1*. Capillary ECs were identified using *CD36*, *CA4*, and *FABP4* [48]. Lymphatic ECs were identified using *PROX1*, *LYVE* [48], *TIMP1*, *CCL21* [32], and *C4orf48*. Proliferating ECs were identified using *SFN* [49], *DMKN*, *KLF5*, *KRT1*, and *KRT14*. Post-capillary venules (PCVs) were identified using *IRF1*, *SELE*, *LRG1*, and *ERG2* [47]. Pre-venular capillaries (PVCs) were identified with *SOX17*, *PLAUR*, *KLHL21*, *GJA1*, *LITAF*, *ICAM1* [47], and *ANXA1*. Finally, we identified the pericyte/endothelial cell cluster using *RGS5* [50], *LGALS1*, and *APOE*. The highest presence of each gene signature was used to identify these subclusters (Figure 2B).

When proportions of LS and healthy cells were investigated within each subcluster based on the DEG profiles between the clusters, the classical endothelial sub-lineages of venous, arterial and lymphatic ECs were found to be mostly prevalent in LS skin. Venous EC had the highest proportion of LS cells at 75%, and pericytes/EC had the lowest proportion of LS cells at about 50% (Figure 2C). DEG analysis of the LS transcriptome compared with healthy controls revealed several upregulated genes of interest that relate to fibrotic and inflammatory processes (Figure 2D). *XIST* was one of the highest differentially upregulated in LS across all endothelial cell subtypes. In addition, *PECAM1* which is a known endothelial marker, *IGFBP4*, *CD34*, *IGLC2*, *HBA2*, *ADAM15*, *HBA2*, *TIP2*, and *COL4A2* were also upregulated in the LS endothelial dataset overall compared with healthy controls. *COL2* was found to be significantly downregulated in LS compared with healthy controls. Interestingly, previous studies have shown that *COL2* expression can vary, being downregulated during inflammation and during advanced fibrosis in certain disease conditions. Appendix A show more information regarding this Seurat object’s metadata.

### 2.3. JAG/NOTCH Signaling between Arterial Endothelial Cells and Capillary Endothelial Cells

Two subclusters, spatially adjacent to each other in the UMAP plot, indicating nearest neighbor clustering with a semi-shared phenotype and arterial and capillary EC (Figure 2A), were the next area of focus based on their significantly upregulated JAG and NOTCH pathways compared with healthy controls. Our dataset contains cell counts of 1141 arterial ECs and 2423 capillary ECs (Appendix A). The volcano plots to visualize DEGs between LS and healthy in the different EC subclusters demonstrate *JAG1* and *NOTCH4* to be significantly upregulated in arterial and capillary clusters, respectively (Figure 3A,B,E). *NOTCH* signaling is often involved and associated with fibrosis development, myofibroblast formation, and the epithelial-mesenchymal transition, which are also important in the disease progression in different forms of scleroderma. We further investigated these pathways due to their dual importance in inflammatory diseases [51].

NicheNet analysis, directed from arterial EC as a sender to all cell clusters as receivers, demonstrated a high potential for arterial cells to communicate via *JAG* signaling (Figure 3D and Appendix A). Top ligands include *ITGB1* with receptors *ENG, CD46*, and *JAM2*. *JAG2* is also a top upregulated ligand with predicted receptors including *NOTCH1*, *NOTCH4* (upregulated in capillary EC), and *CD99*. *SERPING1* is predicted to interact with receptor SELE. Nichenet output indicated a high potential for this ligand to interact with receptor *NOTCH4*, which is highly expressed in capillary ECs (Appendix A).

Our data revealed crosstalk between arterial ECs and capillary ECs via *JAG1* and *NOTCH4* signaling (Appendix A and Figure 3A,B). The *NOTCH* ligand and receptor display EGF-like repeats, relevant as *EGF* genes are often upregulated within our LS dataset. Dot plots of EC vs. all cells showed that the *JAG1* signaling originates from endothelial cells, specifically arterial, interacting with smooth muscle cells and keratinocytes. Additional interactions include *RPS19* to B cells; *CD99*, *FN1* and *CXCL12* to fibroblasts; *SELE* intrinsically to endothelial cells; and *IL6* and *CCL2* to smooth muscle cells (Figure 3C). Together, this suggests that arterial endothelial cells promote JAG/NOTCH pathway intrinsically within arterial and capillary endothelial cells, and extrinsically with smooth muscle cells and keratinocytes.

Further NicheNet analysis of ECs showed *JAG1* ligand signals sent from endothelial cells (Figure 3C) and received by target gene *KRT10* in follicular keratinocytes via NOTCH interactions (Figure 3D). *JAG* is also predicted to interact with *MT2A*, involved in IFN-stimulated pathways [52]. Arterial ECs as signal sender in NicheNet analysis indicated communication with other endothelial cells (Appendix A). Both of these clusters of (arterial and capillary) ECs are predicted to be responsible for the *JAG/NOTCH* signaling internally within endothelial cells, and externally with smooth muscle cells and keratinocytes (Figure 3C). *HES1*, a known *NOTCH* target gene, is heavily expressed in LS patients compared with healthy controls, particularly in keratinocytes [53] (Figure 3F).

### 2.4. Extrinsic SELE Signaling Originating from Endothelial Cells with Predicted Receptor CD44 and Extracellular Communications and Crosstalk between JAG/NOTCH Pathway

The *SELE* gene emerged as one of the most upregulated genes across almost all endothelial cell–subtype DEG lists. *SELE* is observed in both LS and healthy cells (Figure 4E) and is almost exclusively expressed in endothelial cells (Figure 4B). However, *SELE* signaling, according to NicheNet and CellChat communication analyses, is entirely dependent on disease state (LS vs. healthy) (Figure 4A, Appendix A), highlighting its potential role in disease propagation in LS. Cytokine-stimulated endothelial cells express *SELE*, facilitating the migration of blood leukocytes to the sites of inflammation by mediating the adhesion of cells to vascular lining [54]. *SELE* is highly expressed in pre-venular capillaries, post-capillary venules, pericytes, and proliferating ECs. Interestingly, *SELE* expression was higher in healthy proliferating ECs than in LS cells but was still found to be a significant signaling ligand (Appendix A). Conversely, *SELE* expression in healthy pericytes is significantly lower than LS pericytes (Figure 4E). E-selectin, the protein encoded by SELE, is critical in recruiting leukocytes to sites of inflammation and damaged skin during chronic inflammation and autoimmunity [55,56,57,58]. This underscores the importance of further investigation of SELE in our dataset.

CellChat includes *SELE* signaling in its CellChatDB, providing valuable insight into how this gene interacts within our dataset. We found that SELE likely serves as a strong extrinsic signal from endothelial cells to all other cell types in the skin, except for follicular keratinocytes (Figure 4A). This signaling involves *CD44* as a receptor (Figure 4F), which is highly expressed in LS cells across all clusters aside from endothelial cells (Figure 4G). Additionally, intrinsic communications between *CCL14*, *CCL2*, and *ACKR1* are highly probable within endothelial cell subclusters (Figure 4H). CCL was also a significant pathway in our CellChat analysis, and the contributing cell types involved can be found in Appendix A.

*SELE* was not only an important extrinsic signal (Figure 3C) but also a prominent intrinsic ligand as shown in the NicheNet ligand interaction dot plot (Appendix A). It appears as both an intrinsic and extrinsic signaling molecule according to Cellchat in silico modeling. *SELE* was one of the top 20 predicted ligands in our capillary-to-all endothelial cell NicheNet analysis (Appendix A), helping to identify the type of endothelial cell promoting extrinsic SELE signaling. *SELE* is prominently expressed in our LS endothelial dataset, specifically in the pre-venular capillaries and post-capillary venules (Appendix A) and is also predicted to be a significant signaling molecule (Figure 3A). In our ligand-target NicheNet analysis, we observed that *IL-6*, a pro-inflammatory cytokine, and *CX3CL1*, a chemotactic cytokine, are both predicted to target SELE (Appendix A). 

SELE interacts with genes of the NOTCH pathway, such as *MYC* [59]. *MYC* was found to be an upstream regulatory factor in the NOTCH pathway, indicating there may be an overall connection between these two endothelial pathways [59]. *SELE* interactions with other target genes in the LS disease process include *ICAM1*, *CCL2*, *HES1*, *ID3*, *IRF1*, *JUN/D/B*, *SOCS3*, and *CLIC* as noted from this cellular communication analysis. The top predicted receiver of *SELE* in this interaction analysis is *GLG1* (Appendix A), a gene predicted to enable fibroblast growth factor binding (*FGFBP1*) [60]. This suggests that SELE potentially plays a significant role in fibrosis during LS, corroborated by the increased activity of *FGFBP1* by SELE in diseased endothelial cells in NicheNet. 

Additionally, SELE was identified as a ligand involved in signaling from endothelial cells to monocytes/macrophages, T cells, and suprabasal keratinocytes (Figure 4H). This interaction proposed by our CellChat analysis named *CD44* as the predicted receptor/target. The SELE signaling network (Figure 4C) indicates *SELE* signaling from endothelial cells to suprabasal keratinocytes, granular keratinocytes, mast cells, T-cells and monocytes/macrophages demonstrating a strong extrinsic influence on both inflammatory and stromal cells.

Overall, arterial endothelial cells are highly involved in JAG/NOTCH signaling in our dataset. NOTCH1, known to negatively regulate epithelial progenitor cell proliferation, is essential for differentiation. Our capillary and arterial GSEA analyses (Figure 5B,D), demonstrated upregulation of genes related to the p53 pathway, which may promote NOTCH signaling in endothelial cells, leading to interactions with keratinocytes, contributing to LS manifestations. The clear upregulation of *JAG* and the significant presence of the p53 pathway in arterial EC suggest that arterial EC are the primary initiators of this signaling pathway in our dataset.

Arterial EC DEGs (Figure 5A) in LS samples showed upregulated genes like *XIST*, *ITGB4*, *EFNB2*, and *A2M*, which may influence disease progression. *JAG1* and *JAG2* were highly expressed in the arterial subcluster (Figure 3D). Other notable upregulated genes in arterial EC include *ICAM2*, *SEMA3G*, *FBLN2*, *CLIC5*, *KRT19*, *IGF2.1*, and *COL21A1*. Downregulated genes in this cluster, compared with other endothelial clusters, included *ACKR1*, *IGFBP7*, *CCL14*, *SELE*, *IL6*, and *COL15A1* (Figure 3A). GSEA analysis of the LS arterial DEGs indicated involvement in inflammatory pathways including TNFA signaling via NFKB, IL2/STAT5 signaling, P53 pathway, IL6/JAK/STAT3 signaling and KRAS signaling (Figure 5A,B).

Capillary ECs showed upregulated DEGs like *NOTCH4*, *LGALS1*, *COL4A2*, *Cllorf96*, and *CXCL2* (Figure 3C). This subgroup showed similar upregulated LS DEGs to the overall EC LS DEGs, including upregulation of *XIST*, *IGFBP4*, *PLCG2*, *ITGA9*, *TGFBR2*, *RPL37A*, and *PLVAP* (Figure 5C). GSEA overlap analysis on capillary EC LS DEGs indicated involvement in several inflammatory pathways including TNFA signaling via NFKB, KRAS signaling, P53 pathway, IL2/STAT5 signaling, IL6/JAK/STAT3 signaling, and TGFb signaling (similar to arterial cell GSEA). Other significant DEGs in this cluster included *CD36*, *CLU*, *ACKR1*, *NOTCH4*, *ITGA1*, *SELE*, *LGALS1*, *HLA-C*, *FABP5*, and *IL32*, with *SELE* and *NOTCH4* highly significant in disease pathways (Figure 5D). These DEGs also overlapped in hypoxia, apoptosis, IFG response, epithelial mesenchymal transition, adipogenesis, inflammatory response, myogenesis, and angiogenesis (Figure 5D).

In addition to our single-cell dataset, we have included *JAG* and *NOTCH* expression as seen in an LS patient’s affected biopsy for proof of concept. Appendix A shows the biopsy stained with H&E, highlighting areas of inflammation. The blue/purple indicates lymphoplasmacytic cellular infiltrate (circled). Spatial plotting of JAG and NOTCH related genes showed high expression in the area of lymphocytic infiltrate (Appendix A).

### 2.5. XIST Expression Higher in LS across All Endothelial Cell Subtypes

We observed high expression of *XIST* (X-inactive specific transcript) in LS endothelial cells overall when comparing LS with healthy controls (Figure 2D), including LS arterial and capillary EC subclusters (Figure 5A,C). *XIST* was the most significantly upregulated gene in LS endothelial cells (*p*-value 1 × 10^−300^) and showed an increased log fold change compared with healthy controls (Figure 2D). Although XIST is not directly implicated in LS, it has known roles in female-predominant diseases. XIST plays a critical role in endothelial immune function and apoptosis, with downregulation leading to healthier functioning cells [59]. Understanding the role of XIST in endothelial cells of LS patients could provide insights into the immune response, especially given its significance in SSc and other autoimmune disease that predominately affect females [60]. Given that *XIST* was the most upregulated gene in our LS dataset and that LS is primarily a female autoimmune disorder, we considered it of particular interest in our study. This suggests that XIST may play a pivotal role in disease progression in female LS patients and could also have underlying roles in male LS. To ensure that the high expression of *XIST* was not due to gender bias, we analyzed our dataset and found a well-balanced female:male ratio in gene expression among all endothelial cells. This confirms that the prominence of XIST is due to high expression in diseased samples, both male and female patients (Figure 5E,F). XIST appears to function as an intrinsic signaling factor within endothelial cells. We also find that *XIST* is not localized to any endothelial subcluster but is observed to be upregulated in all LS samples (Figure 5G,H).

### 2.6. Potential Roles of IL-33 and IL6 in LS Endothelial Cells

NicheNet analysis on proliferating endothelial cells (ECs) to all ECs identified *IL-33* as one of its top predicted ligands, interacting with IL1RL1 in our dataset (Appendix A). IL-33, an alarmin cytokine, plays an anti-inflammatory role during tissue injury and also promotes ECM remodeling via T reg and M2 macrophages [61,62], mediating intricate intercellular communication between immune cells to regulate cutaneous immune responses. Epidermal keratinocytes are the primary producers of IL-33, followed by dermal fibroblasts and macrophages, while endothelial cells have been shown to express *IL-33* and IL-33 receptors [62,63].

In our endothelial dataset, *IL-33* is strikingly expressed in endothelial cells compared with all cell types in the skin (Appendix A) and is upregulated as compared with healthy controls in across many endothelial subclusters (Appendix A), including venous, capillary, post-capillary venules, and arterial endothelial cells. IL-33 also consistently appears among the top 20 predicted ligands in various NicheNet analyses, including arterial EC as senders (Appendix A) and the capillary EC as senders (Appendix A). IL-33 is also likely to be communicating with venous ECs via signaling from proliferating ECs and arterial ECs (Figure 6). IL-33 was also found to interact with *IL-6*, *ICAM1*, *SOCS3*, and other inflammatory response gene (Appendix A), likely stimulating inflammation and propagating myofibroblast-expressing ST2, also upregulated in our dataset.

Another inflammatory cytokine previously linked to SSc, *IL-6*, was a highly expressed in ligand-mediating communication from LS endothelial cells to LS smooth muscle cells in our dataset (Figure 3F). IL-6 is produced by inflammatory and immune responses in autoimmune disease by T cells, B cells, monocytes, and fibroblasts and is a cytokine found to be elevated in the peripheral circulation of LS patients [16], though their function in endothelial cells is less documented. NicheNet inference predicts that IL-6 targets many genes relevant to fibrosis including *CCL2*, *ICAM1*, *SOCS3*, *JUND*, *NAMPT*, *ICAM1*, *CD34*, and *FABP4* (Appendix A).

## 3. Discussion

Abnormal endothelial cell responses are considered part of the initiating event in systemic sclerosis (SSc), intertwined with immune activation and stromal activation of the skin fibroblasts. This concept is not as well studied in localized scleroderma (LS), a clinical phenotype of scleroderma with predominant skin and underlying tissue involvement. Both entities share histopathological signatures, notably a peri-vascular lymphoplasmacytic infiltrate in the background of a dense collagen extracellular matrix. We investigated endothelial cell (EC) subsets in LS to predict interaction of ECs with other immune cells and fibroblasts and explored their potential involvement in LS disease propagation.

In this study, we used single-cell transcriptomic analysis of LS tissues from a diverse patient population (pediatric, adults, multiple races and sexes) to characterize endothelial cell clusters in LS, compare them to matched healthy controls, and study their dysregulated pathways to elucidate the role of EC in pathogenesis and their similarity to the disease processes in SSc. VEGFA, a prototypical angiogenic factor in ECs found to be elevated in many pulmonary vascular diseases, has been implicated in disease propagation in SSc [64,65,66]. However, this depth of understanding is underexplored in LS, with few studies analyzing LS using scRNAseq [67]. Our group has previously demonstrated the inflammatory role of fibroblasts in LS, showing their strong in-silico communication and staining co-localization with T cells and macrophages in the skin [30,68,69]. In this study, we provide an unprecedented view of endothelial cell heterogeneity in LS, identifying eight populations (subclusters) with robust gene expression markers per subcluster, reflecting discrete functions. Almost all EC populations were more prevalent in LS compared with healthy control. Among them, four subclusters overlapped with gene expression identified in adult SSc EC subtypes, arterial (*HEY1/FBLN*), capillary (*CD36/CA4/FABP4*), venous (*ACKR1*), and lymphatic (*PROX/LYVE*) ECs, supporting some overlap with SSc [48]. At the same time, our pre-venular capillaries and post-capillary venules contain key upregulated genes like *PLAUR* and *SELE,* respectively, that were dissimilar to those seen in SSc, signifying localized scleroderma has independent signatures and pathways distinct from SSc.

While SSc is characterized by widespread multiorgan vascular disease and extensive EC injury, our data indicate upregulation and co-communication of *JAG/NOTCH* pathway signaling, a known precursor of fibrosis that facilitates myofibroblast formation and epithelial-mesenchymal transition, suggesting a pre-fibrosis mechanism in LS. Dysregulation of these key genes in the *NOTCH/JAG* pathway, specifically in arterial and capillary ECs, as well as genes related to apoptotic signatures, draws similarity of disease pathogenesis as observed in SSc skin [70] and may be important in LS disease progression.

Activation of the *NOTCH* signaling pathway, often observed in angiogenesis and fibrotic diseases, leads to phenotypic, morphological, and functional changes in the epithelium, contributing to damage in the epithelial layer and disease progression. *NOTCH* signaling impacts cell differentiation, proliferation, survival, and apoptosis [71], and is involved in fibrosis, myofibroblast formation, and epithelial-mesenchymal transition [72,73,74,75], all of which play an instrumental role in SSc [76]. Furthermore, inhibition of NOTCH signaling prevents the development of fibrosis in bleomycin-induced fibrosis and in Tsk-1 mouse models [73,77]. Overexpression of *JAG1*, a known ligand of NOTCH1, observed in fibroblasts in SSc skin leads to overexpression of collagen [73]. We observe similar NOTCH/JAG pathway upregulation in LS cohort endothelial cells, corroborating that LS may share aspects of SSc pathogenesis despite its ‘non-systemic’ nature; this altogether supports our hypothesis that NOTCH signaling via JAG with smooth muscle plays an important role in the pathogenesis of LS.

SELE signaling was observed to be an important pathway in our LS endothelial cell subset. SELE is a cytokine-induced vascular adhesion molecule responsible for multiple inflammatory signaling cascades in diseased endothelial cells [78]. Studies in SSc lungs have found that *SELE* (and *ICAM1*) are found to be more highly expressed in diseased cells compared with healthy controls [79] and its inhibition could lead to an increase in successful therapeutic outcomes [78]. We observe similar gene expression in our LS skin analysis. SELE likely facilitates crosstalk between endothelial cells and keratinocytes, fibroblasts, macrophages, and T-cells, with the target gene being *CD44* (Figure 4H). We predict that *SELE* is a promising marker gene for locating activated and injured endothelial cells in LS. Another molecule in this pathway, GLG1, is recognized SELE ligand that initiates leukocyte assembly in the vascular endothelium [80], and its increase may promote fibrosis by activating the fibroblast inflammatory response. Together, the enhanced SELE pathway in LS endothelial cells may lead to inflammation and migration of leukocytes to areas of endothelial injury and promote peri-vascular immune infiltrate observed histologically in LS, ultimately promoting disease. Two inflammatory cytokines, IL-6 and IL-33, were consistently represented throughout the LS endothelial cell analyses, including overall EC, subtypes, and cell communication software analyses, and were both linked to SSc disease promotion. IL-33 has been demonstrated to serologically correlate with the degree of skin thickness, pulmonary fibrosis, and early disease state [81] [23,24,25]. In SSc mouse models and in SSc ex vivo skin models, IL-33 endothelial cell responses have been shown to demonstrate both inflammation and fibrosis [82]. IL-33 upregulation is likely to cause an overactive immune response even when no infection is present, hence may be driving autoimmune disease, making it essential to understand its pathway (activation/inhibition) for potential therapeutic intervention in localized scleroderma. Tozorakimab, which is an anti-IL-33 monoclonal antibody, has demonstrated success in inhibiting IL-33 by disrupting the *RAGE/EGFR* signaling pathway through the inhibition of ST2 [83]. Tozorakimab’s ability to reduce IL-33–induced inflammation makes it a promising novel treatment for LS patients with high levels of IL-33 activity.

Another cytokine of interest in our dataset, IL-6, already has a therapeutic inhibitor named Tocilizumab (anti–IL-6 receptor antibody) commonly used in autoimmune disease and FDA-approved for interstitial lung disease in systemic sclerosis [84]. IL-6 is known to be upregulated in the peripheral blood of patients with LS [16]. It promotes B-cell proliferation, regulates fibroblast activity, and initiates a healing response caused by inflammation [85]. IL-6 contributes to disease progression by promoting collagen production and reducing collagenase production [86]. IL-6 is a known cause of inflammatory activation [26], vascular leakage, leukocyte recruitment, and impediment of blood flow in endothelial cells [27,28]. This suggests that the increased IL-6 signal in our LS dataset may mediate signaling in damaged vascular endothelial cells, further contributing to the progression of LS.

IL-6 has also been involved in three different signaling methods: trans signaling, cluster signaling, and classical signaling. In classical signaling, receptors for IL-6 can be generated by *ADAM17* and *ADAM10,* two genes we observed to be upregulated in LS endothelial cells. Among these methods, trans-signaling has a profound impact on disease state and is an important cascade to understand in LS [87]. In our dataset, IL6 from the pre-venular capillary EC are predicted to signal with smooth muscle cells, likely playing additional roles in the process of fibrosis of LS. Additionally, inhibition of classic and trans IL-6 signaling can also be accomplished by targeting glycoprotein 130–associated Janus kinases (JAKs) [87]. The prevalence of the multiple down and upstream signaling genes involved in IL-6 signaling in our LS dataset highlights IL-6 as a key player in disease progression. Moving forward, understanding and therapeutically targeting different methods of IL-6 signaling in LS endothelial cells will be crucial given the past successful treatments of IL-6 with Sirukumab and Tocilizumab in SSc [88]. Overall, this implies that IL-6 has significant influence in LS disease progression, promoting both the pro-inflammatory and fibrotic signaling pathways.

Finally, the most upregulated gene in our data, *XIST*, is involved in X-linked silencing in cells by regulating epigenetic processes and has been associated with several autoimmune diseases that are female predominant, such as multiple sclerosis and systemic sclerosis [89]. XIST also plays a role in the oxidative stress response, inflammation, and regulating endothelial dysfunction [90]. It is significant in SSc regulatory pathways, including those involving IL-6, CCL2, and SERPINE1 [89], all of which are upregulated in our LS data. Additionally, XIST participates in hypoxia-induced angiogenesis [91], and hypoxia was denoted as one of the top GSEA overlap pathways in our PVC LS DEG analysis (Appendix A). The high upregulation in XIST in LS endothelial cells suggests a strong role in disease progression, particularly in vascular dysregulation. This prominence indicates a cell type-specific role for XIST in endothelial cells, as seen in other diseases related to vascular angiogenesis. XIST also regulates KLF4- found to be highly expressed in LS PVCs and is a crucial gene for endothelial cell regulation and proliferation [92].

By studying endothelial cells and their interaction in LS, we have identified four key pathways that likely play significant roles in LS pathogenesis. These pathways involve both intrinsic and extrinsic signaling between endothelial cells themselves, as well as with other cell types. The arterial and capillary ECs intrinsically fuel the *JAG/NOTCH* pathway and interact with fibroblasts through GLG1 and FGFB1, thereby increasing fibrosis. SELE, produced by the damaged endothelial cells, interacts with keratinocytes and fibroblasts, driving T-cells to the injury site and serving as a promising marker for injured endothelial cells in LS, indicative of perivascular infiltration and inflammation and neutrophil recruitment. IL-33 and IL-6 also play vital roles in the disease process of LS. *IL-33*, found in our LS dataset, interacts with IL-6 and triggers fibrosis through interaction with ST2, which stimulates myofibroblasts to further produce these ligands. Increased IL-6 also promotes collagen production, fibrosis, and mediates signaling in damaged vascular endothelial cells, further contributing to LS progression. IL-6 from pre-venular capillary ECs is predicted to signal with smooth muscle cells, with upregulation in the *JAK/STAT* pathway likely playing additional roles in fibrosis. The prevalence of the multiple down and upstream signaling genes involved in IL-6 signaling underscores the role of IL-6 as a key player in disease progression. Finally, *XIST*, involved in X-linked silencing via epigenetic regulation, is upregulated in LS and plays a role in the oxidative stress response, inflammation, and regulation of endothelial dysfunction. The upregulation of XIST, which is also seen in other female-predominant autoimmune diseases, likely plays a central role in vascular dysregulation and LS progression.

### Limitations

Despite promising observations suggesting that endothelial cells play a crucial role in LS disease pathogenesis, more detailed insights are required to better define the pathogenesis and progression of disease. Our scRNA-seq analysis had an adequate size overall, but relatively small sample size per LS subtype (linear trunk, linear head, generalized morphea, etc.) as well as between pediatric- and adult-onset LS. Expanding the cohort would allow to identify potential subcluster EC differences between patients with these clinical subtypes. Additionally, this study utilized a cross-sectional design, but analysis and collection of longitudinal samples of the same patients could be highly beneficial. Longitudinal data would allow us to study the changes in endothelial subclusters and changes in DEGs over time. This approach would enhance our understanding of how the disease state effects the transcriptome and how these insights could inform treatment options. Though this study provides valuable insight and clues into the possible pathogenesis of LS from a single-cell analysis standpoint, this data will need to be verified in future using IHC, PCR, and Western blots and backed up with functional assays.

## 4. Materials and Methods

### 4.1. Human-Patient Skin Samples

Each study participant consented to two 4-mm area punch biopsies of the affected area in the case of LS patients and forearm in the case of controls. One of the biopsies was used for running single-cell sequencing (scRNA seq) and the second was paraffin-embedded for sectioning later. These samples were collected from research participants in the National Registry of Childhood Onset Scleroderma (NRCOS) (the University of Pittsburgh #STUDY19080297), Connective Tissue Disease (CTD) Registry (University of Pittsburgh, #STUDY19090054), and Morphea in Adults and Children (MAC) Registry (University of Texas Southwestern, #STU112010-028) cohorts. All the LS patients enrolled in the study were either grouped into pediatric onset if disease onset of <18 years of age or adult onset if onset ≥18 years of age. Healthy controls were age and sex match whenever possible and was availed through IRB designed for discard tissue collected (University of Pittsburgh, #STU19070023).

### 4.2. Single-Cell RNA Sequencing

Fresh or CryoStor^®^ CS10 (BioLife Solutions^®^, Bothell, WA, USA) preserved samples were processed, as described in our earlier feasibility publication [29]. These biopsy samples were enzymatically digested (Miltenyi Biotec Whole Skin Dissociation Kit, human Cat# 130-101-540 Miltenyi Biotec^®^, Gaithersburg, MD, USA); Miltenyi gentleMACS Octo Dissociator (Cat# 130-096-427, Miltenyi Biotec^®^, Gaithersburg, MD, USA). The suspended cells were filtered using 70-μm cell strainers and re-suspended in 0.04% BSA+ PBS. Post filtration, the cells were mixed with reverse transcription reagents and loaded into the Chromium instrument (10x Genomics^®^, Pleasanton, CA, USA), a commercial application of Drop-Seq [73]. The Chromium instrument then formed GEMs, which are gel beads in emulsion. GEMs contain a gel bead, scaffold for an oligonucleotide that is composed of an oligo-dT section for priming reverse transcription, and barcodes for each cell (10x Genomics^®^) and each transcript (unique molecular identifier, UMI) [74]. Approximately 4000 cells were loaded per sample. V1, V2, and V3 single-cell chemistries were used in accordance with the manufacturer’s protocol (10x Genomics). The reaction mixture was briefly removed from the Chromium instrument, and reverse transcription was performed. The emulsion was then broken using a recovery agent, and following Dynabead and SPRI clean up, cDNAs were amplified by 11–12 cycles of PCR (C1000, Bio-Rad, Hercules, CA, USA). cDNAs were sheared (Covaris, Woburn, MA, USA) into ~200 bp length. DNA fragment ends were repaired and A-tailed, and adaptors were ligated. The library was quantified using a KAPA Universal Library Quantification Kit KK4824 (Roche Diagnostics Corporation, Indianapolis, IN, USA) and further characterized for cDNA length on a bioanalyzer using a high-sensitivity DNA kit(Illumina, San Diego, CA, USA). Libraries were sequenced (~200 million reads/sample), using the Illumina NextSeq-500 platform (Illumina, San Diego, CA, USA).

### 4.3. Spatial Transcriptomics

Visium Spatial Gene Expression (10x Genomics, Pleasanton, CA, USA) was run on formalin-fixed paraffin-embedded (FFPE) skin biopsy from the affected area of an LS patient. 5 µm sections were taken from FFPE and analyzed on Visium CytAssist [93] for FFPE tissue slide cassette with 6.5 mm × 6.5 mm capture area. The sequencing data and the Hematoxylin and Eosin (H&E) stained image were merged and aligned by the spatial coordinate of the slide.

### 4.4. Data Preprocessing and Bioinformatics Analyses

Sequencing reads were examined by quality metrics, and transcripts were mapped to reference human genome (GRCh38). Cell Ranger (10x Genomics, Pleasanton, CA, USA) was used to assign reads to particular cells according to their barcode. Data from the study will be deposited on NCBI Gene Expression Omnibus (GSE number GSE264508, release date of 21 March 2025; additional GSE number to be determined, pending).

Data analyses were performed using R (version 4.2), specifically the Seurat 4.3 package for normalization of gene expression and identification, and visualization of cell populations [75,76]. The data was subset by the parameters of nFeature_RNA > 150 and <2700, percent.mt < 40, percent. ribo < 50 and nCount_RNA < 8000. Genes were filtered according to the Filter Genes function, with a min. Value of 0.5 and min. cells of 100. Principal component analysis (PCA) was performed on the highly variable genes, and the Harmony [77] package was used to integrate the dataset, removing variation by sample (library_id) before cells were clustered using a smart local moving algorithm (SLM) [78] and visualized by UMAP [79]. Clustering was performed on our data using FindNeighbors with the dimension parameter set to 15 and using the “harmony” reduction. Clusters were calculated using the FindNeighbors function with a resolution set to 0.7 (https://satijalab.org/seurat/articles/get_started_v5_new, (accessed on 1 March 2024)). Code is adapted from the saltija lab.

Differential gene expression was performed between sample types, LS compared with healthy among the cell clusters and subclusters using the Libra [94] package’s edgeR based “pseudobulk” methodology. *p*-values were calculated and adjusted according to the Benjamini Hochberg method, and differential expression was reported as a log-fold change (log FC). The package Enhanced Volcano was used to visualize differentially expressed genes through volcano plots.

### 4.5. Cell-Communication Analysis via NicheNet and CellChat

NicheNet [95], a Seurat compatible package, was used to investigate the cell-to-cell communication. This utilizes bona fide ligand-target interaction matrices, gene regulatory interactions, as well as signal transduction to make predictions based on our input data. We chose a sender cell type as well as a receiver cell type(s), and the output provides predictions on the ligands involved in the interaction, the cell types involved in interaction, as well as the target genes and the receptors involved in the same communication network. These findings can then be backed up by plotting where the gene signatures lie in the given dataset.

Similar to NicheNet, CellChat is a R package that utilizes our Seurat object containing our scRNA seq data and does cellular interaction analysis. CellChat utilizes a manually curated database of literature-supported ligand-receptor interactions. Their human database contains 1939 validated molecular interactions. To utilize the CellChat pipeline, we adapted the code posted in their GitHub [96].

## 5. Conclusions

In-depth analysis of endothelial cells within our dataset of 44 samples reveals significant involvement of these cells in several relevant inflammatory and pro-fibrotic pathways leading to phenotypic changes in LS patients. Clustering of EC defined eight unique subclusters, four of which overlap in gene expression of those identified in SSc endothelial cell studies [48], including venous ECs, capillary ECs, arterial ECs, and venous ECs. While there are similarities to SSc, we also discovered new EC subtypes, gene signatures, and pathways specific to LS.

Arterial and Capillary clusters, identified by markers similar to those in SSc, may be of greatest importance in LS disease progression. These cell types fuel the JAG/NOTCH pathway, a strong intrinsic pathway within endothelial cells, which also act extrinsically with fibroblasts to potentially increase fibrotic response through GLG1 and FGFB1. NOTCH signaling is also upregulated in endothelial cells, promoting fibrotic and inflammatory responses in fibroblasts due to upregulation of FBGF1 through GLG1. JAG and NOTCH genes, localized in areas of inflammatory peri-vascular infiltrate (Appendix A), are likely key signaling players in angiogenesis and fibrosis, thus aiding the disease process.

In addition to these strong intrinsic signals, we identified extrinsic cellular interactions between endothelial cells and keratinocytes, smooth muscle cells, fibroblasts, T cells, macrophages, and B cells. SELE signaling from endothelial cells to fibroblasts may increase in inflammatory responses by inducing FBGF1. SELE, produced by damaged endothelial cells, interacts with keratinocytes and fibroblasts, driving T-cells to the site of injury and serving as a promising marker for injured endothelial cells, evidenced by the perivascular inflammatory infiltrate in LS histology.

Two cytokines, IL-33 and IL-6, also play vital roles in LS disease progression. IL-33 interacts with IL-6 and triggers fibrosis through ST2, which triggers and stimulates myofibroblasts. Increased IL-6 promotes collagen production and fibrosis and mediates signaling in damaged vascular endothelial cells, further contributing to the progression of LS. IL-6 from pre-venular capillary ECs is predicted to signal with smooth muscle cells, with upregulation in the JAK/STAT pathway likely playing additional roles in fibrosis. The prevalence of multiple down and upstream signaling genes involved in IL-6 signaling emphasizes the role of IL-6 as a key player in disease progression.

Finally, XIST, involved in X-linked silencing via epigenetic regulation, is upregulated in LS and plays a role in oxidative stress response, inflammation, and endothelial dysfunction. The upregulation of XIST, seen in other female-predominant autoimmune diseases, likely plays a crucial role in vascular dysregulation and LS progression. This study identifies these four main molecular pathways of interest, which will provide a broader understanding of landscape of LS pathogenesis and help target these pathways to design therapeutic interventions, starting with the investigation of functional assays as the next step.

## Figures and Tables

**Figure 1 ijms-25-10473-f001:**
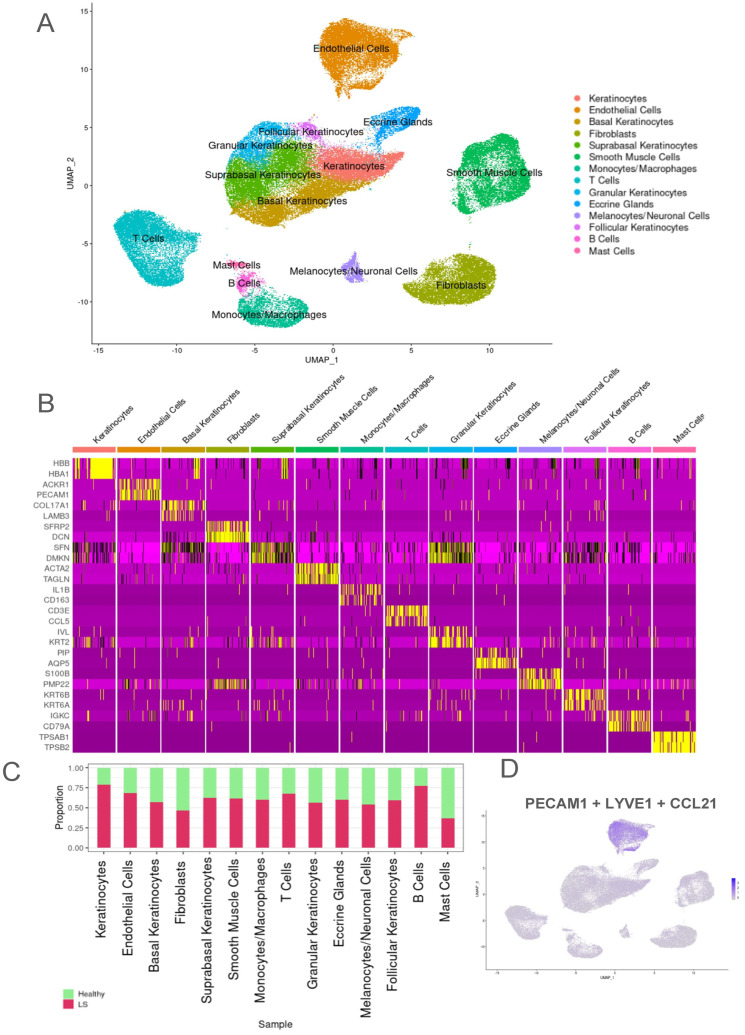
Transcriptomic evaluation of endothelial cells show higher proportions in LS samples. (**A**) UMAP of all cells from 44 total samples of LS (27) and healthy (17) samples with a total of 108,239 cells of which 40,715 LS and 67,524 healthy cells clustered into 14 main cell types. (**B**) Bar graph visualizing the percentage/ proportions of each cell type between LS and healthy cells in the entire population. (**C**) Heat map used for identifying the different cell clusters using the top 4 genes in each cluster. (**D**) UMAP feature plot showing composite canonical endothelial cell markers, *PECAM1*, *CCL21*, and *LYVEI*, used to identify the endothelial cell populations.

**Figure 2 ijms-25-10473-f002:**
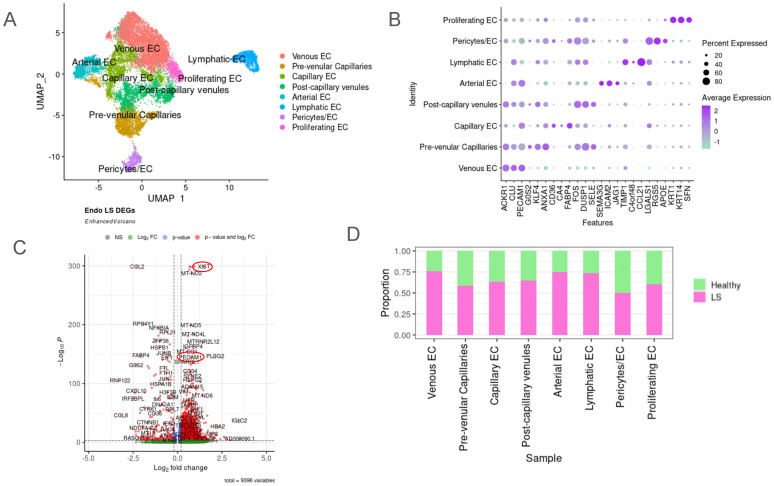
Unique clusters identified in endothelial cells (EC). (**A**) A total of 16,766 cells were identified as ECs and were clustered into eight main subsets as seen in the UMAP. These included venous EC, pre-venular capillaries, capillary EC, post-capillary venules, arterial EC, lymphatic EC, pericytes/EC and proliferating EC. (**B**) Dot plot representing the three main identifying genes for each subcluster of EC. (**C**) Most EC subclusters had a higher proportion of LS cells, except for pericytes/EC. (**D**) Volcano plot of dysregulated genes in the overall endothelial cell subset compared with healthy endothelial cells, with attention to *XIST* and *PECAM1* (circled) as significantly upregulated in LS endothelial cells.

**Figure 3 ijms-25-10473-f003:**
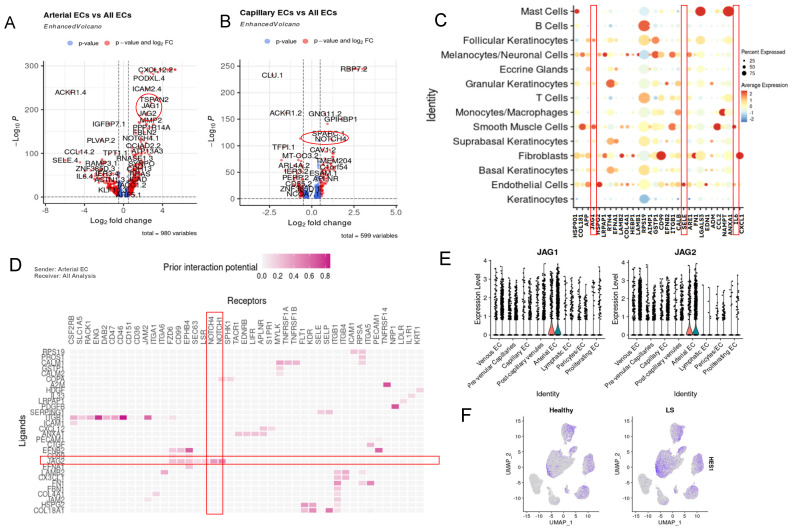
Upregulation of *JAG/NOTCH* Signaling between Arterial and Capillary Endothelial Cells. (**A**) Volcano plot displaying dysregulated DEGs in the arterial EC cluster vs. all other endothelial cell subclusters showing upregulation in *JAG1* and *JAG2* (circled in red). (**B**) Volcano plot displaying dysregulated DEGs in the capillary EC cluster vs. all other endothelial cell subclusters shows upregulation in *NOTCH4* (circled in red), *LGALS1*, *COL4A2*, *C11orf96* and *CXCL2*. (**C**) NicheNet interaction dotplot of the top 20 predicted ligands ECs as sender and all cells as receiver. Highlighted are genes of interest, including *JAG1, SELE* and *IL6* (boxed in red). (**D**) NicheNet ligand-target interaction potential plot displaying the interaction potential between top predicted ligands involved in arterial endothelial cell extrinsic signaling to all other endothelial subsets (weighted by LS vs. Healthy)—projecting JAG2 (boxed in red horizontal) as a top predicted ligand (*y*-axis) with predicted target genes *NOTCH1* and *NOTCH4* (*x*-axis) (boxed in red vertical). (**E**) The violin plot of *JAG1* and *JAG2* expression amongst endothelial cells for all EC subclusters. (**F**) UMAP displaying showing where *HES1*, an important gene in the JAG/NOTCH pathway, localizes in our all-cells dataset split by healthy vs. LS.

**Figure 4 ijms-25-10473-f004:**
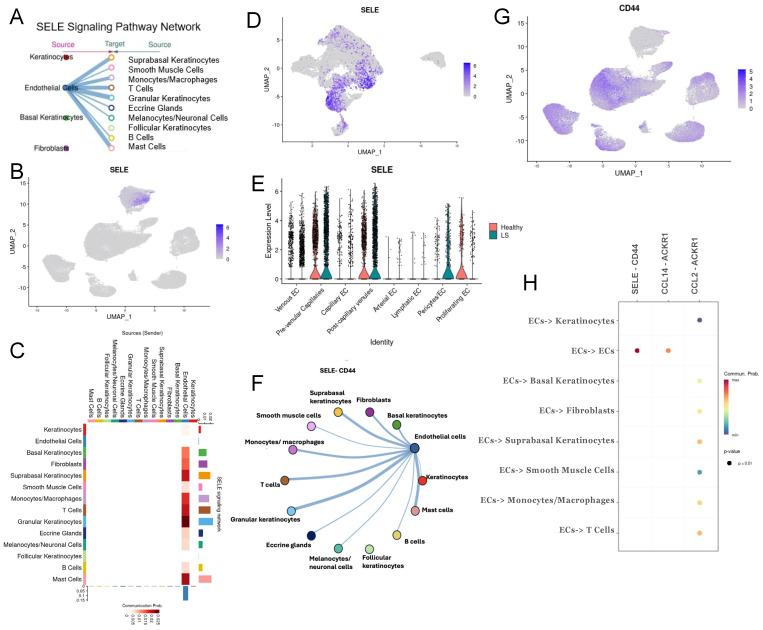
Pathway showing extrinsic *SELE* signaling in ECs. (**A**) CellChat paracrine signaling pathway network of *SELE* in our dataset with cell types listed on the left being SELE expressing cell types—which consists of only endothelial cells. The interactions include signals from endothelial cells to all cells aside from follicular keratinocytes. (**B**) UMAP displays the clusters where *SELE* is mostly relevant in all cells from our data and shows its presence in endothelial cells. (**C**) Signaling matrix using CellChat displays the SELE signaling network with the magnitude of interaction from endothelial cells to other interacting cell type, with the strongest to granular keratinocytes, T cells, mast cells and suprabasal keratinocytes. (**D**) UMAP displays the location of *SELE* in the endothelial cells subclusters and shows its expression in pre-venular capillaries, post-capillary venules and proliferating endothelial cells. (**E**) Violin plot showing the expression level of *SELE* in each cluster in endothelial cell subset split by health and is more prevalent in healthy population. (**F**) Circos plot showing interaction pathways of *SELE* to *CD44*. (**G**) UMAP projection of the cell types that express the SELE receptor *CD44*, which includes all cell types aside from endothelial cells. (**H**) CellChat’s predicted communication pathways shown that stems from endothelial cells and all other cells. This is included as SELE-CD44 with a positive communication probability for endothelial cells to T cells, monocytes/macrophages and suprabasal keratinocytes.

**Figure 5 ijms-25-10473-f005:**
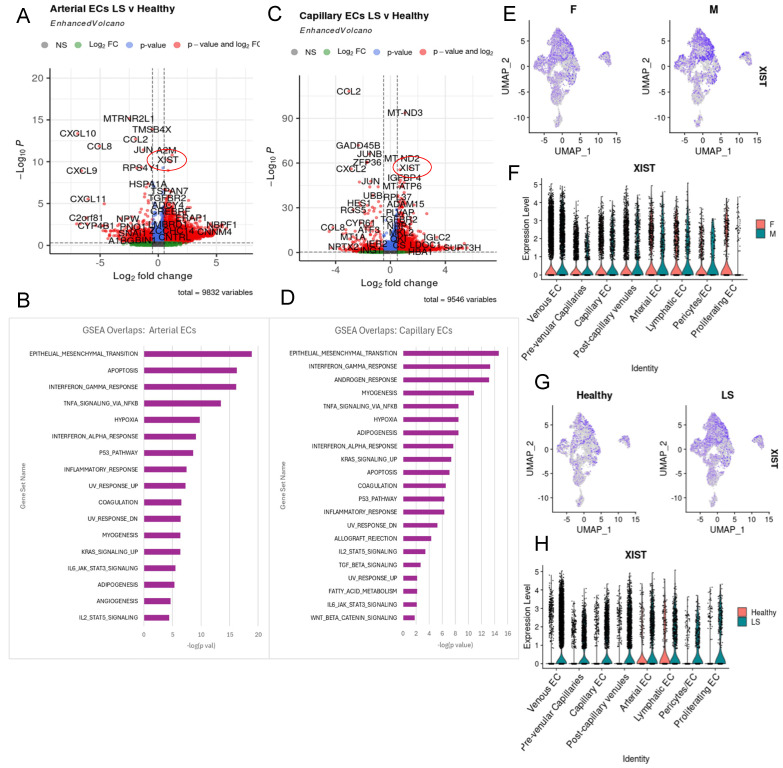
XIST in LS endothelial cells. (**A**) Volcano plot showing the top LS vs. healthy DEGs for the arterial cluster. (**B**) Top 20 GSEA pathway overlaps ranked by *p*-value based on the top 200 upregulated DEGs in the arterial LS. (**C**) Volcano plot showing the top LS vs. healthy DEGs for the capillary cluster. (**D**) Top 20 GSEA pathway overlaps when run on the top 200 upregulated DEGs in LS capillary DEGs. (**E**,**F**) Displays the proportion of male vs. female cells that express in endothelial cells in a feature plot and a volcano plot respectively. (**G**) Shows the expression of XIST via feature plot in our endothelial cell subset split by health. (**H**) Shows a volcano plot of XIST expression per cluster in our endothelial cell subset split by health with clear prominence in LS cells.

**Figure 6 ijms-25-10473-f006:**
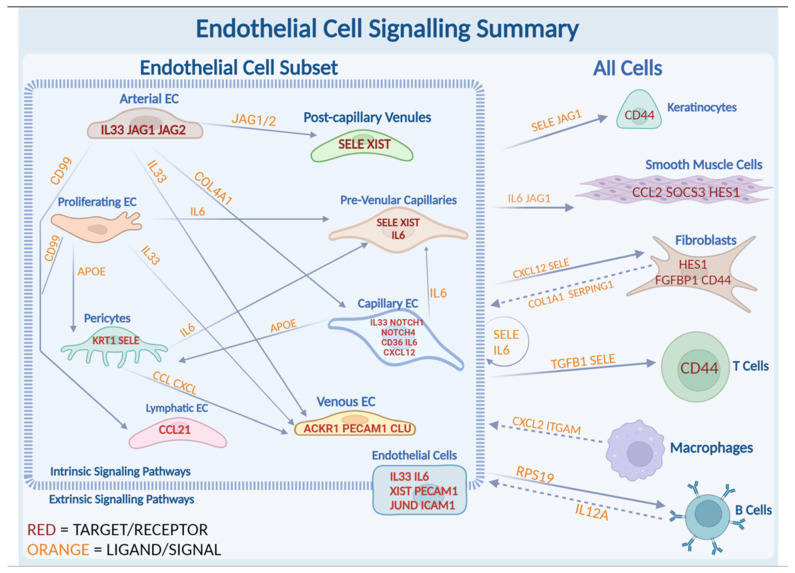
Intrinsic and extrinsic signaling pathways in endothelial cells within themselves or with other cell types upregulated in LS disease process. Interactions amongst different subclusters of endothelial cells and between endothelial cells and all other cell types in our dataset are depicted. Within the endothelial cell subset, the most significant interactions stemming from arterial EC, proliferating EC, pericytes, pre-venular capillaries, lymphatic EC, venous EC and capillary EC are displayed. Additionally, signaling from endothelial cells to keratinocytes, smooth muscle cells, fibroblasts, T cells, macrophages and B cells are illustrated to the right side of the figure. Genes colored orange were identified in CellChat or NicheNet analyses and represent ligands, and genes in red were found in NicheNet or feature plots as receptors or target genes.

## Data Availability

Data from the study will be deposited on NCBI Gene Expression Omnibus as fastq files (raw) and processed data.

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
