# Peer review of "Characterization of Endothelial Cell Subclusters in Localized Scleroderma Skin with Single-Cell RNA Sequencing Identifies NOTCH Signaling Pathway"

_ijms, 2024, doi:10.3390/ijms251910473_

Round 1

Reviewer 1 Report

Comments and Suggestions for Authors

The authors examined the genetic alteration of endothelial cells of human LS using single cell RNAseq to gain an insight into the role of endothelial cells in the pathophysiology of LS. They analyzed the single cell RNAseq with various informatic tools to identify the molecular pathways involved in LS. Overall the data is rich and provides clues for the understanding of LS. However, the data needs verification through IHC, PCR or western blot in the future. Following are some points to consider for further process.

1.     “Transcriptomic Evaluation of Endothelial Cells from Both LS and Healthy Control Samples Identified LS Samples to be Predominant Over Healthy Cells in the Endothelial Cell Population” predominant in what aspect?

2.     Fig 1 B is incorrigible due to small font size.

3.     LS samples show higher cell numbers in most of the cell types. Is the cellularity higher in the LS tissue samples? If not, the cell numbers shall be normalized to avoid bias.

4.     “Endothelial cell conical markers”->” Endothelial cell canonical markers”

5.     In Fig 2D, COL2 was significantly down-regulated in LS endothelial cells, which is unexpected considering fibrotic nature of LS. However, this point was not discussed. Significantly down-regulated genes need to be addressed.

6.     XIST is a non-coding RNA that acts as a major effector of the X-inactivation process. It is hard to understand its involvement in LS. This shall be discussed in more details.

Comments on the Quality of English Language

Some parts are unclear and need to be clarified.

Author Response

Please see attached detailed responses.

Reviewer 2 Report

Comments and Suggestions for Authors

It is an interesting study for the field. The authors demonstrate through sequencing studies the relevance of endothelial cells and their signaling pathways in the pathogenesis of scleroderma skin, finding the Notch pathway to be the most important. However, the presentation of the results and the low quality of the images in the manuscript do not allow for smooth comprehension of the findings. I also recommend that the molecules mentioned in the text be included in the figures or the analysis. For example, JAG1, as the resolution of the images shown is low, I was unable to appreciate this molecule in your analysis. Additionally, the discussion is very lengthy, and, in some parts, it seems like a description of the results again. I highly recommend restructuring the presentation of the data and providing better quality figures, as well as showing complete figures and not like the one shown in Figure 3C.

Author Response

Please see attached detailed response

Round 2

Reviewer 2 Report

Comments and Suggestions for Authors

I'm sorry but I cannot review the manuscript given the poor quality of the figures. 

I attach a screenshot so you can appreciate the quality of the figures.

Author Response

Comment 1:

The reviewer showed concern that the manuscript figures were poor quality.

Reply to comment 1:

Thank you for that feedback. We have carefully addressed the concerns and worked on every single figure to increase the depth, size, clarity and pixels. We have increased font sizes wherever possible and made significant modifications to enhance the clarity and presentation of our results. The corresponding revisions are embedded in the resubmitted manuscript and track changes applied.

We appreciate your suggestions, as they have helped us to improve the manuscript, and we hope that the revised version now meets the expectations of clarity and quality.